# Multiagent Intratumoral Immunotherapy Can Be Effective in A20 Lymphoma Clearance and Generation of Systemic T Cell Immunity

**DOI:** 10.3390/cancers15071951

**Published:** 2023-03-24

**Authors:** Kristy E. Gilman, Andrew P. Matiatos, Megan J. Cracchiolo, Amanda G. Moon, Dan W. Davini, Richard J. Simpson, Emmanuel Katsanis

**Affiliations:** 1Department of Pediatrics, University of Arizona, Tucson, AZ 85721, USA; 2Department of Cell and Molecular Biology, University of Arizona, Tucson, AZ 85721, USA; 3Department of Immunobiology, University of Arizona, Tucson, AZ 85721, USA; 4School of Nutritional Sciences and Wellness, University of Arizona, Tucson, AZ 85721, USA; 5The University of Arizona Cancer Center, University of Arizona, Tucson, AZ 85721, USA; 6Department of Medicine, University of Arizona, Tucson, AZ 85721, USA; 7Department of Pathology, University of Arizona, Tucson, AZ 85721, USA

**Keywords:** intratumoral, immunotherapy, cancer, checkpoint inhibitors, GM-CSF, combination therapy

## Abstract

**Simple Summary:**

Immunotherapy has shown promise clinically, but resistance to therapies remains an issue. Using novel combinations of immunotherapies may help overcome these barriers. Here, we tested combinations of immunotherapy agents that were administered intratumorally using a mouse model of lymphoma. We show that combination treatment with multiple immune-modulating agents allowed for greater tumor control and induced tumor-specific long-term immunological memory that was T cell dependent. Combination treated animals showed modest alterations in immune cell subsets in lymphoid tissues and had increased immune infiltration to tumors 24 h after treatment. Future studies should evaluate the impact of other immunotherapy combinations on a variety of tumor types.

**Abstract:**

The use of immunotherapies has shown promise against selective human cancers. Identifying novel combinations of innate and adaptive immune cell-activating agents that can work synergistically to suppress tumor growth and provide additional protection against resistance or recurrence is critical. The A20 murine lymphoma model was used to evaluate the effect of various combination immunotherapies administered intratumorally. We show that single-modality treatment with Poly(I:C) or GM-CSF-secreting allogeneic cells only modestly controls tumor growth, whereas when given together there is an improved benefit, with 50% of animals clearing tumors and surviving long-term. Neither heat nor irradiation of GM-CSF-secreting cells enhanced the response over use of live cells. The use of a TIM-3 inhibitory antibody and an OX40 agonist in combination with Poly(I:C) allowed for improved tumor control, with 90% of animals clearing tumors with or without a combination of GM-CSF-secreting cells. Across all treatment groups, mice rejecting their primary A20 tumors were immune to subsequent challenge with A20, and this longstanding immunity was T-cell dependent. The results herein support the use of combinations of innate and adaptive immune activating agents for immunotherapy against lymphoma and should be investigated in other cancer types.

## 1. Introduction

Immune evasion is a well-recognized hallmark of cancer [1,2]. The major goal of immunotherapy is to overcome immune escape and allow for antigen recognition, clearance, and generation of long-term memory against cancer cells. The field of cancer immunotherapy has grown rapidly with the clinical application of checkpoint inhibitors (CPIs), particularly programmed death-1/programmed death ligand-1 (PD-1/PD-L1) and cytotoxic T-lymphocyte-associated protein-4 (CTLA-4) [3,4,5]. The use of CPIs in cancer therapy has been met with great success for certain tumors, but malignant cells can develop resistance to many of these immunotherapies. Therefore, finding novel combinations of immune activating agents may more effectively overcome these resistance mechanisms.

The enhancement of immune responses with non-self-adjuvants was first attempted as a cancer therapy in the 1950s with the use of Bacillus Calmette Guerin (BCG) [6], a vaccine containing live, attenuated Mycobacterium bovis that was initially generated for tuberculosis by Calmette and Guerin [7]. Since then, many novel compounds have been discovered and evaluated in cancer immunotherapy. One such compound is Polyinosinic:polycytidylic acid (Poly(I:C)), a synthetically derived double-stranded RNA (dsRNA) that activates multiple innate immune receptors to direct pro-inflammatory responses [8]. Poly(I:C) has been used as a vaccine adjuvant to enhance immune responses and has also shown promise as a combination therapy for cancer therapy [9,10]. Poly(I:C) has shown promising results in the setting of solid tumor immunotherapy that led to initiation of a Phase II clinical trial (NCT02423863) [11], but has not yet been applied clinically in the context of lymphoma.

Checkpoint inhibition (CPI) has become a major research focus for cancer immunotherapy since the discovery of PD-1, PD-L1 and CTLA-4. T cell immunoglobulin-3 (TIM-3) is a less well-characterized CPI that has seen increased interest recently as a potential replacement or combinatory CPI with PD-1/PD-L1 and/or CTLA-4 [12,13]. Conversely, OX40 is a T-cell-activating receptor found on T cells that binds to the OX40 ligand (OX40L) on antigen-presenting cells to induce survival, proliferation, and long-term T-cell memory [14]. Both TIM-3 inhibition and OX40 agonism have shown promise in preclinical and clinical models in recent years when combined with other checkpoint inhibitors for cancer immunotherapy [14,15,16]. The use of immunotherapies for lymphoma has been highly successful and there are now numerous options for clinicians to consider as treatment, including CPIs, chimeric antigen receptor T cells (CAR-T), and bi-specific T-cell engagers (BITE) [17,18]. The selected treatment will depend on numerous factors, such as the stage of the tumor, the expression of certain antigens, and exposure to previous therapies.

Historically, cancer immune therapies—including immune checkpoint inhibitors—have been administered systemically to patients, which can lead to side effects induced by off-target responses of the therapy. Local or intratumoral administration of agents has gained significant interest in recent years [15,19,20,21,22]. Local administration of therapies has many advantages, including benefits of lower drug doses, increased bioactivity at the target site, lower risk of off-target effects, and reduced overall side effects [23]. In this study, we evaluated the use of intratumoral administration of a variety of combinations of immune-activating agents, including allogeneic GM-CSF-secreting cells, Poly(I:C), OX40 agonism and/or TIM-3 inhibition on A20 lymphoma control and long-term survival of mice [24,25,26]. Combination therapies using Poly(I:C), OX40 agonist and TIM-3 inhibitor with and without the addition of allogeneic Neuro2a/GM-CSF-secreting cells induced complete tumor regression in 90% of mice that led to improved survival and tumor-specific immunological memory.

## 2. Materials and Methods

Cell culture. A20 (ATCC no. TIB-208) cells were grown in RPMI-1640 (Hyclone, no. 16750-070, Logan, UT, USA) with 10% fetal bovine serum (FBS, Millipore, no. TMS-013-B, lot VP1705185, Burlington, MA, USA), 5 mL Sodium Pyruvate (100X solution, Hyclone, no. SH30239.01), 5 mL minimum essential medium (MEM) amino acids (50X solution, 11130-051), and 5 mL penicillin/streptomycin (pen/strep 100 U/mL, Gibco, no. 15140-122, Grand Island, NY, USA). B16 cells were grown in DMEM with 10% FBS and pen/strep. Neuro-2a cells (ATCC no. CCL-131) were previously transduced to secrete murine granulocyte-monocyte colony-stimulating factor (GM-CSF) [27], generating Neuro-2a/GM-CSF cells. Neuro-2a/GM-CSF cells were grown in DMEM (Corning, no. 10-013-CV, Corning, NY, USA) with 10% FBS, 5 mL MEM amino acids, 50 μM β-mercaptoethanol (Gibco, no. 21985-023), and 5 mL pen/strep. NIH/3T3 fibroblast cells (ATCC, no. CRL-1658) that were previously transduced to secrete GM-CSF [28], generating 3T3/GM-CSF cells, were cultured in DMEM, with 10% calf bovine serum (CBS, ATCC, no. 30-2030) and 4 mM L-Glutamine (Gibco, 25030-081). Cells were allowed to grow to confluence and split every 2–3 days. Cells were passaged at least twice before using for experiments to ensure adequate growth would be reached, but were not passaged more than 15 times total.

Mice. Female BALB/cJ mice (stock no. 000651) and C57BL/6J mice (stock no. 000664) were purchased from Jackson Labs (Bar Harbor, ME, USA). Mice were kept in a specific pathogen-free environment, had ad libitum access to food and water and were kept on 12-h light/dark cycles. All experiments were approved by the University of Arizona Institutional Animal Care and Use Committee (IACUC).

Tumor models. Prior to inoculation, cells were washed three times in 20 mL of sterile PBS, then resuspended at a final concentration of 10^7^ cells/mL. A20 and B16 tumors were implanted subcutaneously on the right flank of syngeneic BALB/cJ and C57BL/6J mice, respectively, at 1 × 10^6^ cells per mouse in 100 μL total volume of PBS. A20 tumors were allowed to grow for 12 days prior to randomization. B16 tumors were allowed to grow for 3 days prior to randomization. Tumors were sized on the day of randomization and grouped so that each treatment had an equal distribution of tumor sizes at the start. Tumors were sized with calipers 2–3 times per week and animals were removed from studies after reaching > 5000 mm^3^. Tumor size was determined by taking a length (L) and width (W) measurement and calculating volume using the following equation: L × W^2^ × (π/6). Mice were monitored daily for survival. Moribund mice or those with highly ulcerated tumors were removed from studies. For secondary tumor challenge studies, on day 70 following initial tumor challenge, tumor-free mice received an additional injection of 1 × 10^6^ tumor cells subcutaneously on the left flank. Naïve mice that rejected the tumor without therapeutic intervention were not included in analyses, as the success of treatment could not be evaluated without initial tumor formation.

Intratumoral injection of allogeneic GM-CSF-secreting cells with or without Poly(I:C). On the day of randomization, mice received the first of 8 intratumoral injections of indicated treatments. Poly(I:C) (Invivogen, no. NC0465247, San Diego, CA, USA) was prepared in saline per the manufacturers’ instructions to a concentration of 1 mg/mL. Neuro-2a/GM-CSF or 3T3/GM-CSF cells were washed three times in 20 mL of sterile PBS. Where indicated, Poly(I:C) and the appropriate cell type were mixed for a final intratumoral injection of 50 μg Poly(I:C) and 5 × 10^5^ live GM-CSF-secreting cells (either Neuro-2a/GM-CSF or 3T3/GM-CSF). In single-agent-treated tumors, the opposing volume was sterile PBS to ensure injections were equal in volume across all groups.

Intratumoral immunotherapy administration. On the day of randomization, mice received the first of 3 injections of indicated treatments. All injection volumes were equal across groups. In vivo purified OX40-agonizing monoclonal antibodies (clone: OX-86, no. BE0031) and TIM-3 inhibitory monoclonal antibodies (clone: RMT3-23, no. BE0115) were purchased from BioXCell (Lebanon, NH, USA). OX40 was diluted in sterile PBS to reach a concentration of 1 mg/mL, which was then added to injection mixtures at a concentration of 10 μg per injection. TIM-3 was added at 100 μg per injection, where appropriate. Poly(I:C) and Neuro-2a/GM-CSF cells were prepared as described above with 50 μg and 5 × 10^5^ live cells added per injection, respectively. Vehicle control injections were 100 μL sterile PBS. Poly(I:C)+ OX40 and Poly(I:C)+ TIM-3 groups received all 3 injections of the above-described concentrations of each component. Both the Poly(I:C)+OX40+TIM-3 group and the Neuro-2a/GM-CSF+ Poly(I:C)+ OX40+ TIM-3 group received varying mixtures of agonizing/inhibitory antibodies over the 3 doses: dose 1 was OX40 alone, dose 2 was both OX40 and TIM-3, and dose 3 was only TIM-3. All 3 doses contained Poly(I:C) and Neuro-2a/GM-CSF cells, where appropriate.

In vivo T-cell depletions. At day 120 following primary tumor challenge, surviving tumor-free mice and select naïve mice were depleted of CD4 (BioXCell, clone GK1.5, no.BE0003-1) and CD8 (BioXCell, clone 2.43, no. BE0061) T cells with 100 μg of both depletion antibodies administered intraperitoneally (IP). Mice were challenged with 1 × 10^6^ A20 tumor cells for a third time the next day on the upper right flank. Depletion antibodies (100 μg of both CD4 and CD8) were administered again 3 days later and then weekly until animals were removed from the study. To confirm depletion of T cell subsets, mice were bled approximately every 2 weeks and blood was prepared for flow cytometric analysis. Red blood cells (RBCs) were lysed for 15 min with 1X lysis buffer (BD, no. BDB555899), quenched with media and resuspended in flow buffer (0.5% FBS in PBS). Then, samples were incubated in anti-murine CD16/CD32 (eBiosciences (San Diego, CA, USA), clone 93, no. 501129520) for 15 min to block non-specific Fc receptor binding. Then, antibodies against murine- CD45 (Biolegend (San Diego, CA, USA), clone 30-F11, no. 103122), CD3 (Biolegend, clone 145-2C11, no. 100338), CD4 (Biolegend, clone RM4-4, no. 116023) and CD8 (Invitrogen, clone CTCD8a, no. MA517599) were added to samples and incubated on ice and in the dark for 30–40 min. Samples were washed with 1 mL flow buffer, resuspended in 200 μL of PBS, run on an LSR Fortessa flow cytometer (BD Biosciences, San Jose, CA, USA) and analyzed using FlowJo software (BD Biosciences).

RNA isolation, cDNA synthesis, and RT-PCR. Twenty-four hours following the second intratumoral injection, spleens were collected and snap-frozen with liquid nitrogen. Then, RNA was extracted with the RNeasy mini kit (Qiagen, no. 74104, Hilden, Germany) per the manufacturer’s instructions with the RNase-free DNase kit (Qiagen no. 79254). The RNA concentration was determined with a nanodrop and diluted to 400 ng/μL. Then, cDNA was synthesized utilizing the iScript Reverse Transcription Supermix (Bio-Rad, no. 1708841, Hercules, CA, USA) following the manufacturer’s instructions. The following primers were designed using primer blast (NIH) and purchased from integrated DNA technologies (IDT): interferon-γ (IFN-γ, fwd: AGC AAG GCG AAA AAG GAT GC, rev: TCA TTG AAT GCT TGG CGC TG) and S15 ribosomal RNA (S15, fwd: ACT ATT CTG CCC GAG ATG GTG rev: TGC TTT ACG GGC TTG TAG GTG). Primer efficiencies were confirmed prior to running RT-PCR. RT-PCR was run using SYBR green PCR master mix (Applied Biosystems, no. 4364344, Waltham, MA, USA) per kit instructions on a Roche Lightcycler96 (Roche, Basel, Switzerland) and analyzed with the 2-ΔΔCt method.

Immunohistochemistry. Tumors were harvested from mice 24 h following the second intratumoral injection and were put in 10% neutral buffered formalin overnight at 4 °C and then transferred to 70% ethanol. Samples were then dehydrated, cleared, embedded in paraffin, and sectioned at 5 μm at the University of Arizona Tissue Acquisition, Cell and Molecular Analysis Shared Resource. Tissue sections were incubated at 37 °C for 20 min and washed twice in Histo-clear (Fisher, no. 5089990147, Waltham, MA, USA) for 5 min to warm and remove paraffin, respectively. Then, tissues were rehydrated with two 5-min washes in alcohol gradations (100, 95, 70, 50%) and deionized water. Slides were then permeabilized for 15 min in 0.2% Triton X-100 PBS, then washed with PBS. Then, antigen retrieval was done by submerging slides in warm 10 mM citrate buffer (Sigma, no. C9999) for 30 min. Slides were washed 3 times with PBS for 5 min and then blocked for 1 h with 10% goat (Sigma, no. G9023, St. Louis, MO, USA) and 10% donkey (Sigma, no. D9663) serum. Tumor sections were then incubated with rat anti-mouse CD4 IgG (1:100 in 5% BSA PBS, Invitrogen, no. 501129024) and rabbit anti-mouse CD8 IgG (1:400 in 5% BSA PBS, Cell Signaling Technologies, no. 98941S, Danvers, MA, USA) overnight at 4 °C in a humidified chamber. The following day, slides were washed thrice for 10 min in PBS, then incubated in AlexaFluor488-conjugated goat anti-rabbit IgG (1:500 in 5% BSA PBS, Thermo, no. A11008) and AlexaFluor568-conjugated donkey anti-rat IgG (1:250 in 5% BSA PBS Thermo, no. A78946) secondary antibodies for 1 h. Slides were again washed with three 10-min PBS washes, submerged in water for 5 min, then counterstained with DAPI (1 μg/mL) for 3 min. Slides were washed once for 10 min, then coverslips were mounted with Fluoromount G mounting medium (Fisher Sci, no. 5018788) and stored at 4 °C. Slides were imaged on the same day using identical camera settings for all images on a Leica DM5500 fluorescence microscope (Leica Microsystems, Wetzlar, Germany) with a 4-megapixel Pursuit camera (Diagnostic Instruments, Inc., Livingston, UK) at 400× magnification. CD4- and CD8-positive cells were quantified by manually counting the number of positive cells from 5 fields of view per mouse. Total cell numbers were quantified using ImageJ software (NIH) using the automated Analyze Particles option. Graphs depict the average number of CD4- or CD8-positive cells out of the total cell number for each individual mouse in each group.

Tissue harvest and preparation. One day after the second intratumoral injection, spleens, tumors, and tumor-draining and non-draining lymph nodes were collected for flow cytometric analysis of immune populations. Spleens were collected and processed to single-cell suspensions by mincing and filtering through a 100 μm strainer twice. RBCs were lysed in 1X lysis buffer for 1 min at room temperature, quenched with media containing 10% FBS, then washed, counted, and resuspended in flow buffer for subsequent flow cytometry staining. Tumor digestion solution was prepared with Hank’s balanced salt solution (HBSS, with Ca^2+^ and Mg^2+^, Sigma, no. 55037C) containing 0.5 mg/mL of Collagenase D (Roche, no. 11088858001) and 0.5 mg/mL DNase (Roche, no. 10104159001) and warmed to 37 °C. Tumors were harvested and put in gentleMACS C tubes containing digestion solution and dissociated on the “tumor” series of settings. Then, minced tumors were allowed to digest for 40 min at 37 °C on a nutator. Samples were then run through a 100 μm strainer and washed with HBSS (without Ca^2+^ and Mg^2+^, Gibco, no. 14175-095) containing 5% FBS. Cells were counted and resuspended in flow buffer for staining. Lymph nodes were manually dissociated, run through a 100 μm strainer, counted and resuspended in flow buffer for subsequent flow cytometric staining. For evaluation of intracellular cytokines or granzyme B/perforin levels, cells were treated with 10.6 μM Brefeldin A (Sigma, no. B7651-5MG) and 2 μM Monensin (Sigma, no. M5273-500MG) for 2 h to inhibit protein transport before initiation of staining.

Flow cytometry. Single-cell suspensions of each tissue were first blocked with anti-murine CD16/CD32 for 15 min to block non-specific Fc receptor binding. Then, combinations of extracellular antibodies (listed in Appendix A) were added to each sample and incubated in the dark and on ice for 30–40 min. Samples were washed with 1 mL flow buffer and either resuspended in PBS or fixed for 30 min and then permeabilized with the FoxP3 transcription factor staining buffer set (eBiosciences no. 50-112-8857) for subsequent intracellular staining. Intracellular antibodies were added to appropriate samples, incubated for 30–40 min, then washed and resuspended in PBS. Samples were run on an LSR Fortessa flow cytometer and analyzed using FlowJo software (version 10).

Statistics. Statistical analysis was done using GraphPad Prism 9 software (La Jolla, CA, USA). Survival data was analyzed using the Log rank test. Differences between >2 groups were analyzed using a one-way analysis of variance (ANOVA) with Tukey’s post-hoc test for multiple comparisons or Dunnett’s post-hoc test when comparing to a control group, indicated in figure legends. A *p*-value of < 0.05 was considered statistically significant, with more significant values denoted by numbers of symbols: * <0.05, ** <0.01, *** <0.001, **** <0.0001.

## 3. Results

Injection of Poly(I:C) with or without allogeneic GM-CSF-secreting tumor cells leads to tumor regression and improves survival of mice. We first wanted to investigate whether intratumoral administration of live allogeneic, MHC-disparate cells secreting GM-CSF would sufficiently induce an immune response to allow for control of tumor growth. This hypothesis was evaluated with and without co-injection of the synthetic double-stranded RNA, Poly(I:C). The Neuro-2a/GM-CSF cell line secretes about 5.4-fold higher GM-CSF than the 3T3/GM-CSF lines, secreting roughly 5400 pg/mL versus 1000 pg/mL per 500,000 cells in a 24-h period; therefore, both cell lines were used to evaluate a possible dose effect of GM-CSF. Neither of the GM-CSF-secreting cell lines alone, injected intratumorally, was sufficient to induce complete tumor regression; however, the 3T3/GM-CSF cell line modestly improved survival compared to the PBS (*p* < 0.05), while the Neuro-2a/GM-CSF did not (Figure 1A,B). Poly(I:C) administration improved survival and slowed tumor growth compared to PBS, with 3/11 mice showing complete tumor regression (Figure 1A,B, *p* < 0.01). Co-injection of both GM-CSF-secreting cells with Poly(I:C) led to improved tumor control and survival compared to mice receiving only PBS injections, with 4/10 mice receiving 3T3/GM-CSF cells and 5/10 mice receiving Neuro-2a/GM-CSF cells in conjunction with Poly(I:C) showing complete tumor regression and maintaining long-term survival (Figure 1A,B, *p* < 0.001). Neither group of GM-CSF-secreting cells combined with Poly(I:C) had significantly improved survival when compared to mice treated with Poly(I:C) alone. This experiment shows that Poly(I:C) modestly improves tumor control when given alone or in conjunction with GM-CSF-secreting cells.

We next wanted to evaluate the potential tumor-specific immunological memory of surviving mice. Tumor-free mice were re-challenged with the same tumor 70 days following the initial challenge in the opposing flank, indicated by the black arrows on graphs. Mice that were tumor free upon rechallenge remained tumor free, whereas all naïve mice injected concurrently as controls grew tumors (Figure 1A–C).

Irradiation or heat stress does not augment the immunological response of Poly(I:C) and allogeneic GM-CSF-secreting cell administration. Next, we aimed to assess whether we could augment the effect of our combination intratumoral immunotherapy regimen by stressing the allogeneic GM-CSF-secreting cells prior to injection. Previous studies by us and others have shown that heat-stressing or irradiating cells can lead to release of damage-associated molecular patterns (DAMPs), potential tumor-associated antigens, or upregulation of heat shock proteins such as calreticulin at the plasma membrane, all of which may induce a stronger immune response and allow for greater tumor control, including in a rapidly growing and difficult-to-treat murine tumor model using 12B1 leukemia in syngeneic BALB/c mice [29,30]. Therefore, we assessed whether heat-stressing the Neuro-2a/GM-CSF cells at 43 °C for 120 min or exposing them to 20Gy of X-ray irradiation would enhance the immune response, leading to improved tumor control and survival. Both heat-stressed and irradiated cells combined with Poly(I:C) showed improvements in tumor control and survival when compared to the PBS group (*p* < 0.01), but not over live Neuro-2a/GM-CSF with Poly(I:C) (Figure 2A,B). Therefore, the use of heat stress or irradiation with allogeneic GM-CSF-secreting cells did not further augment the anti-tumor response. Tumor-free mice at day 70 were re-challenged with the same tumor to evaluate long-term immunity, indicated by the black arrow on graphs with naïve mice (Figure 2C) used again as controls. All mice except for one in the irradiated Neuro-2a/GM-CSF +Poly(I:C) group exhibited long-term immunity.

Poly(I:C) and GM-CSF-secreting cell combination induces T cell-mediated immune control of A20 tumors. To evaluate the context of immune-mediated memory, mice that were tumor free following the second A20 challenge were depleted of T cells on day 120 and challenged for a third time the following day. Mice received continuous injections of CD4 and CD8 T-cell-depleting antibodies throughout the experiment and depletion of T cells was confirmed via flow cytometry (Figure 3B,D). Eliminating T cells prior to tumor inoculation in mice with previous immunological memory for A20 tumors led to A20 tumor growth regardless of the treatment regimen (Figure 3), confirming that the responses observed were T-cell dependent. T-cell-depleted mice exhibited enhanced tumor ulceration; thus, many mice in this phase were removed from the study early, before reaching a tumor size of >5000 mm^3^.

Combination immunotherapy allows for improved A20 tumor control. Subsequent studies aimed to reduce the number of injections and improve responses by incorporating immune checkpoint inhibitors/agonists. TIM-3 is a checkpoint inhibitor, of which binding dampens a T-cell response but can be inhibited by the TIM-3 inhibitory antibody. OX40 binding leads to a strong T-cell activation response and can be activated by an OX40 agonizing antibody. For these studies, T-cell targeting with CPIs was utilized either alone or in tandem across three doses of intratumoral immunotherapy that were all in combination with Poly(I:C), with one group also containing Neuro-2a/GM-CSF cells. For groups receiving OX40 and TIM-3 antibodies, OX40 was given with the first and second intratumoral injections and TIM-3 with the second and third doses. The goal of this staggered treatment regimen was to induce a strong T-cell response without leading to T-cell exhaustion, and to limit treatment-related side effects or toxicity. For groups receiving only one CPI, it was administered with all three doses. All treatment regimens slowed tumor growth compared to PBS (Figure 4A,B). Poly(I:C) in combination with TIM-3 allowed for 50% long-term survival (*p* = 0.001 versus PBS), whereas OX40 co-therapy led to 70% survival (Figure 4A,B, *p* = 0.0004 versus PBS). Both groups receiving a combination of Poly(I:C), OX40 and TIM-3 with or without Neuro-2a/GM-CSF administration had 90% long-term survival (Figure 4A,B, *p* < 0.0001 versus PBS) but were not significantly different from each other. Tumor-free mice at day 70 underwent a second tumor challenge (indicated by black arrows on graphs) and showed immunological memory to A20 tumors, as no tumor developed, except for in naïve mice used as controls (Figure 4C). The treatment regimen was also evaluated in another more aggressive tumor model using B16 melanoma in C57BL/6J mice. All treatment regimens modestly improved survival over PBS but did not lead to complete tumor regression (Appendix A, *p* < 0.05). Importantly, no animals exhibited signs or symptoms of treatment-related side effects or toxicity.

Intratumoral immunotherapy does not discernably alter immune subsets in spleens and tumors at acute timepoints post therapy. To attempt to uncover differences in the immune profiles induced by our combination immunotherapy regimens, we evaluated tumor and splenic immune cell subsets 24 h after the second intratumoral dose. This timepoint was selected to allow for evaluation of tumor and spleen in all groups, as many treatment groups had undetectable tumors 24 h after the third intratumoral immunotherapy dose. Only modest alterations were seen at this timepoint. In spleens, there was a slight increase in B cells and decrease in CD4 T cells, in particular naïve CD4 T cells, in the Poly(I:C)+ OX40 +TIM-3 when compared to mice receiving PBS injections (Figure 5A,C,G, *p* < 0.05). Tumor lymphoid subsets were highly variable across treatment groups at this timepoint and did not reach statistical significance (Figure 6). T-cell polarization subsets of helper/cytotoxic type-1, helper/cytotoxic type-2 and helper/cytotoxic type-17 inflammatory T cells were also evaluated in spleens (Appendix A) and tumors (Appendix A) and were not different across groups.

T-cell functionality markers are minimally different in lymphoid tissues and tumors at 24 h post immunotherapy. This work aimed to identify T-cell-mediated mechanistic differences induced by each immunotherapy regimen by evaluating markers of T-cell functionality or cytotoxicity. Interferon-γ (IFN-γ) is a pro-inflammatory cytokine that is upregulated during an immune response and is produced by activated T cells [31]. Thus, we evaluated the level of intracellular IFN-γ in both CD4 and CD8 T cells in spleen, lymph nodes (LN), tumor-draining lymph nodes (TDLN) and tumors from mice in each treatment regimen. Unexpectedly, there was increased IFN-γ+ splenic CD4 T cells in the PBS-injected group as well as the Poly(I:C)+ TIM-3 (Figure 7A, *p* < 0.05). There was also a slight increase of IFN-γ+ CD4 in PBS-injected TDLN when compared to Poly(I:C)+ TIM-3 mice (Figure 7G, *p* < 0.05). IFN-γ+ CD8 T cells were significantly lower in Neuro2a/GM-CSF combination therapy-injected LN compared to all groups except Poly(I:C)+ TIM-3 (Figure 7E, *p* < 0.05) and in TDLN when compared to PBS-injected mice (Figure 7H, *p* < 0.05). Spleen and tumor levels of IFN-γ+ CD8 T cells were not different across groups (Figure 7B,K). We next evaluated CD8 T-cell cytotoxicity by investigating the percentage of Granzyme B+ Perforin+ CD8 T cells, and surprisingly found a reduced percentage in LN of Neuro2a/GM-CSF combination therapy-treated mice (Figure 7F, *p* < 0.05). Tumors from Poly(I:C)+ OX40-treated mice exhibited a significant increase in cytotoxic CD8 T cells (Figure 7L, *p* < 0.05). When looking at each of the aforementioned T-cell markers as a percentage of total CD4 or CD8 T cells, minimal differences were found (Appendix A). We also evaluated the levels of IFN-γ transcripts in whole spleens from treated mice and saw a significant increase in Poly(I:C)+ OX40+ TIM-3-treated animals (Figure 7M, *p* < 0.05) that trended toward significant with Poly(I:C)+ OX40 (*p* = 0.09) or Poly(I:C)+ TIM-3 groups (*p* = 0.09). Lastly, T-cell infiltration of tumors was evaluated via immunohistochemistry. When evaluating the percentage of CD4 and CD8 T cells out of total tumor cell numbers, there was a significant increase of both CD4- and CD8-positive cells in tumors of mice treated with the combination of Poly(I:C)+ OX40+ TIM-3 when compared to those treated with PBS (Figure 7N,O, *p* < 0.05). Evaluation of additional timepoints post treatment may shed more light on T-cell dependent alterations induced by each treatment regimen.

## 4. Discussion

Utilizing non-self-derived adjuvants to mount an immune response has been evaluated for decades and is a major area of interest in cancer vaccine development. The use of allogeneic, MHC-disparate cells or viral/bacterial adjuvants has seen success in preclinical models [20,21,32]. Administration of cells that differ in MHC antigens are known to induce a strong immune response, as differences in MHC complexes are a potent signal to the immune system that non-self-derived cells are present and must be cleared; this phenomenon is clearly reflected in graft-versus-host-disease following hematopoietic cell transplantation (HCT) or in organ rejection following transplantation, but is also harnessed clinically to mediate the graft-versus-leukemia effect post-HCT [25,26,33]. Intratumoral administration of heat-inactivated vaccinia virus Ankara into B16 melanoma and MC38 colon carcinoma tumors successfully induced tumor regression in 100% and 70% of animals, respectively [20]. The use of GM-CSF as a component of cancer vaccines has shown success in mouse models [27,34] and in some cancer treatment regimens [35,36,37]. In recent years, the use of GM-CSF with oncolytic viruses or the development of GM-CSF-producing viruses has shown efficacy in limiting tumor growth. The efficacy of a GM-CSF-producing oncolytic virus against an A20 lymphoma control was shown to be due to an oncoviral-induced release of tumor antigen that induced a CD8+ T-cell response that was able to slow the growth of untreated contralateral tumors, especially when combined with anti-CLTA-4 antibody [21]. The use of oncolytic viruses as a component of lymphoma therapy is in its infancy, but is currently being evaluated in multiple clinical trials (NCT04887025, NCT05387226). A small phase I study (NCT03017820) examining the safety and efficacy of oncolytic viral therapy in patients with relapsed or refractory hematological malignancies has shown impressive results, with remission of lymphoma seen following a single dose of vesicular stomatitis virus, interferon-β with sodium iodide symporter [38]. Likewise, injection of allogeneic fibroblasts transduced to secrete IL-2 or GM-CSF intratumorally into intracerebral breast cancer tumors improved survival over controls [32]. The aforementioned studies were the basis for utilizing allogeneic GM-CSF-secreting cells as a component of our treatment regimen in these studies. Surprisingly, the use of these cells alone was insufficient in mediating A20 lymphoma regression (Figure 1), indicating that injection of allogeneic cells and GM-CSF within evolving tumors does not generate a sufficient immune response to constrain tumor growth. T-cell activation requires a series of coordinated signaling events, including (1) T cell receptor/peptide-MHC binding, (2) T cell co-stimulation and (3) pro-inflammatory cytokine production. Therefore, Poly(I:C) was given alone or added to intratumoral injections to enhance T-cell responses, as Poly(I:C) is an innate immune agonist that should enhance signals 2 and 3 to allow for greater T-cell activation [39]. Poly(I:C) alone modestly slowed tumor growth and improved survival of mice; however, when added to allogeneic GM-CSF-secreting cells, 50% of animals completely cleared tumors, leading to improved survival over controls (Figure 1). Poly(I:C) has shown promise with limiting tumor growth in solid tumors, primarily those of head and neck, which led to initiation of a Phase II clinical trial evaluating its use in a broader context [11]. While Poly(I:C) has not been widely used clinically for treatment of lymphoma, other toll-like receptor agonists, such as CpG, were shown to be safe and capable of inducing an immune response when given intratumorally along with low-dose local irradiation [40,41].

The use of heat stress is known to lead to upregulation of heat shock proteins, such as calreticulin, at the plasma membrane that may further enhance an immune response [29,30]. Irradiated tumor cells have been used as cancer vaccine components for decades [3,27]. Our laboratory has shown that isolation of heat shock proteins from tumor cells that were then used to vaccinate against murine tumors led to a significant reduction of tumor burden [24,28]. We therefore attempted to enhance the tumor-protective effects of the combination of Poly(I:C) and allogeneic GM-CSF-secreting cells by irradiating or heat-stressing them prior to intratumoral injection, which did not improve the immune responses above those observed with injection of live allogeneic GM-CSF-secreting cells (Figure 2).

Checkpoint inhibitors, such as PD-1/L1 and CTLA-4, have been successful at improving tumor control in selective cancers. The use of antibody-mediated T-cell activation with OX40 agonists has gained interest in recent years and has shown promise with preclinical studies in combination with other therapies [14,15,19]. In an aggressive breast cancer model, the combination of OX40 agonism with CpG and an Fc-fused IL-12 protein injected intratumorally into tumors led to tumor regression of both treated and untreated contralateral tumors and significantly improved survival over vehicle-injected mice [19]. Another study evaluated the combination of CpG and OX40 as neoadjuvant therapy and found that tumor control and long-term immunological memory were improved when intratumoral immunotherapy was given 4 days prior to tumor resection; unexpectedly, mice that were given treatment immediately before tumor excision had significantly worse survival [15]. CpG and OX40 therapy was successful at controlling tumor growth in breast and colon carcinoma models and significantly increased the percentage of activated, proliferating T cells. Moreover, the addition of PD-1 inhibition augmented this response [15]. Interestingly, another study compared the success of intratumoral, intramuscular or intravenous administration of Poly(I:C) stabilized with poly-lysine and carboxymethylcellulose, and found that systemic administration (either intramuscular or intravenous) led to increased cytotoxic T-cell infiltration of B16 melanoma tumors in mice [42]. In the current study, we elected to utilize intratumoral injections of all agents used to evaluate the impact on immune infiltration. Combination therapy with Poly(I:C), OX40, and TIM-3 as the primary treatment was able to clear tumors and induce long-lasting immunological memory in an A20 model (Figure 4) and slowed tumor growth in a more aggressive melanoma model (Appendix A). However, only minimal differences were detected in immune cell populations and activation/cytotoxicity markers across treatment regimens at the timepoint studied (Figure 5, Figure 6 and Figure 7, Appendix A). Of note, combination Poly(I:C)+ OX40+ TIM-3 treatment showed a slight reduction in splenic CD4 T cells, specifically naïve CD4 T cells, when compared to PBS-injected mice (Figure 5C,G). Tumor immune cell subsets were not different across groups when looking at percentages of total immune cells (Figure 6); however, percentages of CD4 and CD8 T cells were increased in Poly(I:C)+ OX40+ TIM-3-treated tumors when looking at total tumor cells (Figure 7M,O), with this treatment exhibiting 90% complete tumor regression (Figure 4). These data suggest that this immunotherapy regimen increased the migration of T cells from the spleen to the tumor site to aid in tumor clearance. Evaluation of systemic poly(I:C) administration with intratumoral CPI injections may potentially demonstrate some improvement over the intratumoral route used in this work [42]. Evaluation of additional timepoints over the therapeutic schedule may be more informative for defining mechanistic differences between treatments. Additionally, in vivo monitoring of immune cell migration through lymphoid tissues and into tumors and/or single-cell RNA sequencing of immune cells following treatment could shed light on how each component of the combination therapy mediates responses across immune cell types.

## 5. Conclusions

In conclusion, we show that combination immunotherapy using OX40 agonism, TIM3 inhibition and Poly(I:C) injected intratumorally into A20 lymphoma tumors leads to complete tumor regression and improved long-term survival. Single-modality CPI treatment of either OX40 agonism or TIM3 inhibition with Poly(I:C) improved survival but was less impressive than when given in combination. Addition of GM-CSF-secreting allogeneic cells did not significantly augment the anti-tumor effect. Unexpectedly, immune populations were minimally different across groups, but the combination immunotherapy did show an increase in T-cell infiltration into tumors.

Future studies should attempt to uncover ways to identify tumors that may respond better to certain innate immune activators (CpG, Poly(I:C) or LPS), T-cell activating agents (OX40, 4-1BB or CD28) or those that have enhanced expression of specific checkpoint inhibitors (PD-1, CTLA-4, TIM-3, LAG-3 or VISTA). Additionally, better understanding of how T cells are altered by these therapies at the transcript or signaling level may help identify novel therapeutic targets to help overcome resistance mechanisms that develop with continuous use of immunotherapies.

## Figures and Tables

**Figure 1 cancers-15-01951-f001:**
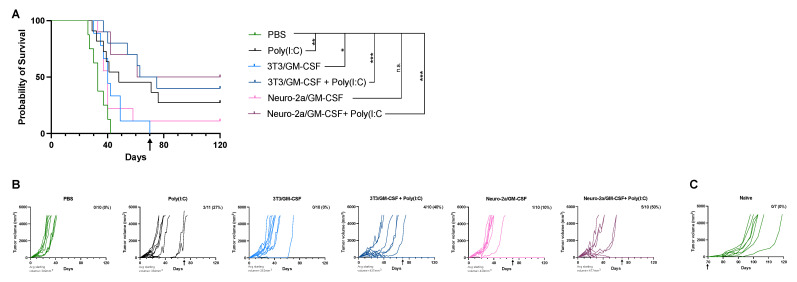
Intratumoral injection of Poly(I:C) with or without the addition of allogeneic GM-CSF-secreting cells into solid A20 tumors leads to tumor regression, prolongs survival, and induces long-term immunity. Tumor-bearing mice were randomized into treatment groups to allow for even distribution of tumor sizes across groups. Mice received indicated treatments twice a week for 4 weeks for a total of 8 intratumoral injections. (**A**) Kaplan-Meier survival curve of mice in each treatment group. Statistical differences were assessed using the Log rank test with the level of significance denoted by the number of asterisks: * *p* < 0.05, ** *p* < 0.01, *** *p* < 0.001, n.s. = not significant. (**B**) A20 lymphoma growth of individual mice in each treatment group. Tumors were sized 2–3 times per week and tumor size was calculated as described in Section 2. Mice were removed from the study when they reached a tumor size of >5000 mm^3^. (**A**,**B**) Tumor-free mice were challenged for a second time with A20 tumor on the opposing flank at day 70, indicated by the black arrow on graphs. (**C**) A20 tumor growth of individual naïve control mice.

**Figure 2 cancers-15-01951-f002:**
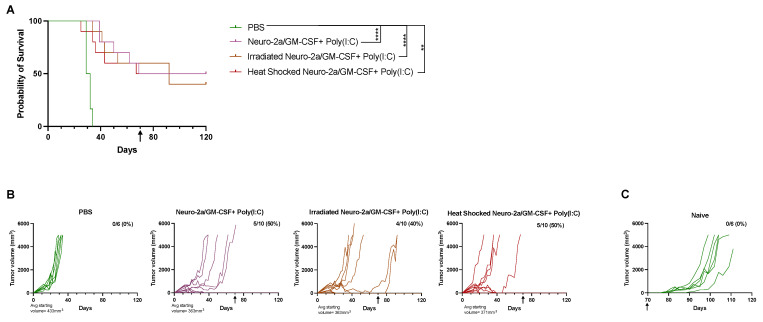
Irradiating or heat-stressing allogeneic GM-CSF-secreting cells prior to co-injecting with Poly(I:C) does not augment A20 tumor control above that seen with live cell injection. Mice bearing A20 tumors were randomized into treatment groups and received intratumoral injections of indicated treatments twice a week for 4 weeks for a total of 8 injections. (**A**) Survival of mice in indicated groups are illustrated with a Kaplan-Meier survival curve. Statistical differences were assessed using the Log rank test with the level of significance denoted by the number of asterisks: ** *p* < 0.01, **** *p* < 0.0001. (**B**) Tumor growth of individual mice. Tumors were sized 2–3 times per week and calculated as described in Section 2. Mice were removed from study once a tumor size of >5000 mm^3^ was reached. (**A**,**B**) Tumor-free mice were rechallenged with A20 tumor on the opposing flank at day 70, indicated by the black arrow on graphs. (**C**) Growth of A20 tumors in naïve mice used as controls.

**Figure 3 cancers-15-01951-f003:**
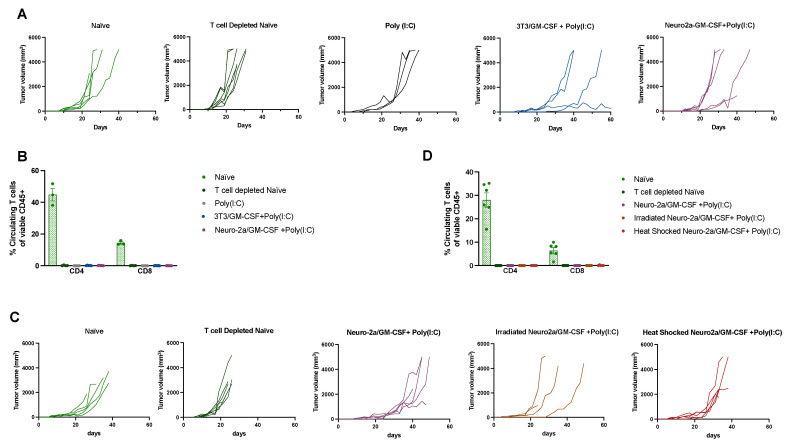
T-cell depletion of mice with prior A20 tumor immunity induced by intratumoral injections of Poly(I:C) and GM-CSF-secreting cells allows for tumor growth upon rechallenge. (**A**–**D**) Tumor-free mice at day 120 following the initial A20 tumor challenge were depleted of T cells as described in Section 2 and were challenged with A20 tumor for a third time on the upper right flank. Tumors were sized 2–3 times per week and mice were removed from study once a tumor size of >5000 mm^3^ was reached, when tumors became highly ulcerated (>50% visible area) or when moribund. Graphs of individual mouse tumor growth in T-cell-depleted mice from (**A**) Figure 1 or (**C**) Figure 2. (**B**,**D**) T-cell depletion was confirmed by evaluating CD4 and CD8 T cell percentages in blood via flow cytometry, as described in Section 2. The percent of circulating T cells per group is illustrated for mice from (**B**) Figure 1 or (**D**) Figure 2, using week 2 depletion data as a representative timepoint.

**Figure 4 cancers-15-01951-f004:**
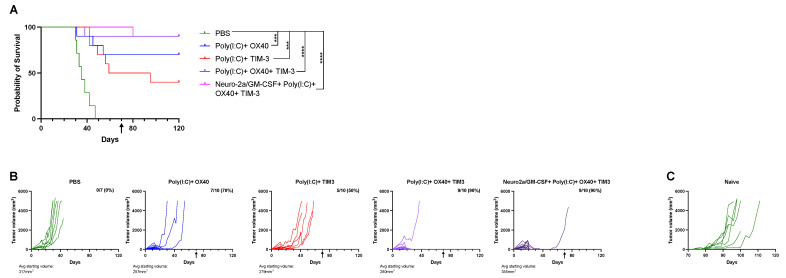
Combination immunotherapy improves tumor control and survival of solid A20 tumor-bearing mice and generates tumor-specific immunological memory. A20 tumor-bearing mice were randomized into treatment groups and mice received 3 intratumoral injections of indicated treatments. (**A**) Kaplan-Meier survival graph of mice in each treatment group. Statistical differences were assessed using the Log rank test with the level of significance denoted by the number of asterisks: *** *p* < 0.001, **** *p* < 0.0001. (**B**) Individual mouse tumor growth illustrated by treatment group. Tumors were sized 2–3 times per week and tumor size was calculated as described in Section 2. Mice were removed from the study once tumors reached a size of >5000 mm^3^. (**A**,**B**) Tumor-free mice were challenged for a second time with A20 tumor on the opposing flank at day 70, indicated by the black arrow on graphs. (**C**) Tumor growth of individual naïve control mice challenged with A20 lymphoma.

**Figure 5 cancers-15-01951-f005:**
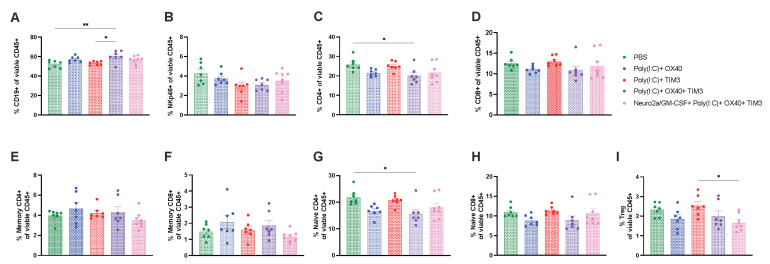
Combination immunotherapy modestly alters splenic immune cell subsets 24 h post treatment. Spleens were harvested 24 h after the second immunotherapy dose and prepared for flow cytometric analysis. Lymphoid cell subsets are illustrated as a percentage of total viable CD45+ cells and gated as follows: (**A**) B cells: CD19+ (**B**) NK cells: NKp46+ (**C**) Total CD4 T cells: CD4+ (**D**) Total CD8 T cells: CD8+ (**E**) Memory CD4 T cells: CD4+CD44+ (**F**) Memory CD8 T cells: CD8+CD44+ (**G**) Naïve CD4 T cells: CD4+CD62L+ (**H**) Naïve CD8 T cells: CD8+CD62L+ (**I**) T regulatory cells: CD4+CD25+FoxP3+. (**A**–**I**) Statistical differences were determined with a one-way ANOVA followed by Tukey’s post-hoc comparisons with the level of significance denoted by the number of asterisks: * *p* < 0.05, ** *p* < 0.01.

**Figure 6 cancers-15-01951-f006:**
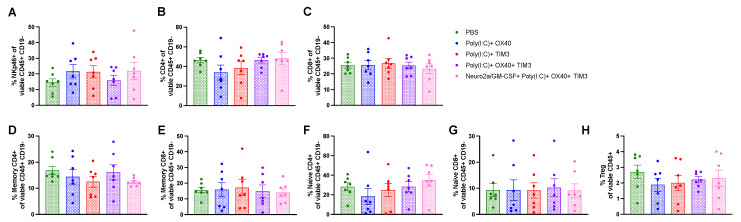
The tumor immune profile is variable 24 h post-intratumoral immunotherapy administration. Tumors were harvested 24 h after the second immunotherapy dose and prepared for flow cytometric analysis as described in Section 2. Lymphoid cell subsets are illustrated as a percentage of total viable CD45+CD19- cells and gated as follows: (**A**) NK cells: NKp46+ (**B**) Total CD4 T cells: CD4+ (**C**) Total CD8 T cells: CD8+ (**D**) Memory CD4 T cells: CD4+CD44+ (**E**) Memory CD8 T cells: CD8+CD44+ (**F**) Naïve CD4 T cells: CD4+CD62L+ (**G**) Naïve CD8 T cells: CD8+CD62L+ (**H**) T regulatory cells: CD4+CD25+FoxP3+. (**A**–**H**) Statistical differences were determined via a one-way ANOVA and Tukey’s post-hoc comparisons.

**Figure 7 cancers-15-01951-f007:**
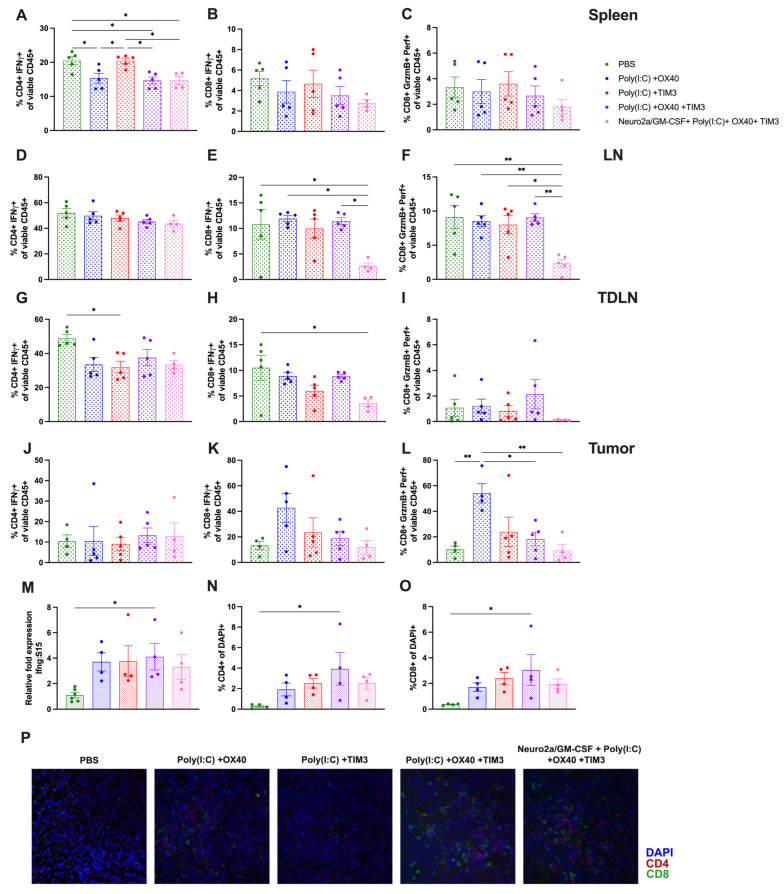
Markers of T-cell functionality are inconsistently altered across lymphoid tissues and in tumors 24 h following immunotherapy. (**A**–**C**) Spleens, (**D**–**F**) lymph nodes (LN), (**G**–**I**) tumor-draining lymph nodes (TDLN), and (**J**–**L**) tumors were harvested 24 h after the second immunotherapy dose and prepared for flow cytometric analysis as described in Section 2. All groups are presented as a percentage of total viable CD45+ cells. Pro-inflammatory T cells were identified via the presence of interferon-γ (IFN-γ) in (**A**,**D**,**G**,**J**) CD4+ T cells or (**B**,**E**,**H**,**K**) CD8+ T cells. (**C**,**F**,**I**,**L**) Active cytotoxic CD8 T cells were identified via co-expression of Granzyme B (GrzmB) and Perforin (Perf). (**M**) Spleens were harvested 24 h after the second immunotherapy dose, RNA was extracted, and cDNA was synthesized as described in Section 2. RT-PCR was performed to assess differences in transcripts of interferon-γ (IFN-γ) across treatment groups. Data are presented as fold change relative to spleens from PBS-injected tumors using S15 ribosomal RNA as an internal control. (**N**–**P**) Tumors were harvested 24 h after the second immunotherapy dose and were fixed, dehydrated, and sectioned for immunohistochemical analysis. Sections were stained as described in Section 2. Total cell numbers and cells positive for (**N**) CD4 or (**O**) CD8 T cells were counted to evaluate the extent of T-cell infiltration of tumors across treatment regimens. (**P**) Representative images of T-cell infiltration into tumors of indicated treatments (400× magnification). (**A**–**O**) Statistical analyses were done using a one-way ANOVA with (**A**–**L**) Tukey’s or (**M**–**O**) Dunnett’s post-hoc test. The level of significance is denoted by the number of asterisks: * *p* < 0.05, ** *p* < 0.01.

## Data Availability

The data that support the findings of this study are available from the corresponding author, E.K., upon reasonable request.

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
