# Peer review of "Multiagent Intratumoral Immunotherapy Can Be Effective in A20 Lymphoma Clearance and Generation of Systemic T Cell Immunity"

_cancers, 2023, doi:10.3390/cancers15071951_

Round 1

Reviewer 1 Report

Kristy E. Gilman and co-authors present a high quality and well-written experimental manuscript focusing on multiagent intratumoral immunotherapy can be effective in tumor clearance and generation of systemic T cell immunity.

Authors evaluated the use of intratumoral administration of a variety of combinations of immune-activating agents including allogeneic GM-CSF secreting cells, Poly(I:C), OX40 agonism and/or TIM-3 inhibition on A20 lymphoma control and long-term survival of mice. Combination therapies using Poly(I:C), OX40 agonist and TIM-3 inhibitor with and without the addition of allogeneic Neuro2a/GM-CSF secreting cells induced complete tumor regression in 90% of mice that led to improved survival and tumor-specific immunological memory.

Authors used the A20 murine lymphoma model to evaluate the effect of various combination immunotherapies administered intratumorally. They showed that single-modality treatment with Poly(I:C) or GM-CSF secreting allogeneic cells only modestly controls tumor growth, whereas when given together, there is an improved benefit, with 50% of animals clearing tumors and surviving long-term. Neither heat nor irradiation of GM-CSF secreting cells enhanced the response over use of live cells. The use of TIM-3 inhibitory antibodies and OX40 agonizing antibodies in combination with Poly(I:C) allowed for improved tumor control, with 90% of animals clearing tumors with or without combination of GM-CSF secreting cells. Across all treatment groups mice rejecting their primary A20 were immune to subsequent challenge with A20 and this longstanding immunity was T-cell dependent. The results support the use of combinations of innate and adaptive immune activating agents for cancer immunotherapy. 

Finally, authors conclude that identifying tumors that may respond better to certain innate immune activators (CpG, Poly(I:C), or LPS), T cell activating agents (OX40, 4-1BB or CD28) or those that have enhanced expression of specific checkpoint inhibitors (PD-1, CTLA-4, TIM-3, LAG-3, VISTA), and continuous monitoring of these markers throughout treatment, may lead to more successful tumor control for patients. 

Overall, the manuscript is highly valuable for the scientific community and should be accepted for publication.

======================

Other comments to authors:

1) Please check for typos throughout the manuscript.

2) Line 388 and further, Figure 7 – with regards to markers of T cell functionality authors are kindly encouraged to cite the following article that extensively covers various aspects focusing on function/dysfunction of T cell-based immunotherapies. DOI: 10.3390/cancers14041078

Author Response

Reviewer 1:

Kristy E. Gilman and co-authors present a high quality and well-written experimental manuscript focusing on multiagent intratumoral immunotherapy can be effective in tumor clearance and generation of systemic T cell immunity.

Authors evaluated the use of intratumoral administration of a variety of combinations of immune-activating agents including allogeneic GM-CSF secreting cells, Poly(I:C), OX40 agonism and/or TIM-3 inhibition on A20 lymphoma control and long-term survival of mice. Combination therapies using Poly(I:C), OX40 agonist and TIM-3 inhibitor with and without the addition of allogeneic Neuro2a/GM-CSF secreting cells induced complete tumor regression in 90% of mice that led to improved survival and tumor-specific immunological memory.

Authors used the A20 murine lymphoma model to evaluate the effect of various combination immunotherapies administered intratumorally. They showed that single-modality treatment with Poly(I:C) or GM-CSF secreting allogeneic cells only modestly controls tumor growth, whereas when given together, there is an improved benefit, with 50% of animals clearing tumors and surviving long-term. Neither heat nor irradiation of GM-CSF secreting cells enhanced the response over use of live cells. The use of TIM-3 inhibitory antibodies and OX40 agonizing antibodies in combination with Poly(I:C) allowed for improved tumor control, with 90% of animals clearing tumors with or without combination of GM-CSF secreting cells. Across all treatment groups mice rejecting their primary A20 were immune to subsequent challenge with A20 and this longstanding immunity was T-cell dependent. The results support the use of combinations of innate and adaptive immune activating agents for cancer immunotherapy. 

Finally, authors conclude that identifying tumors that may respond better to certain innate immune activators (CpG, Poly(I:C), or LPS), T cell activating agents (OX40, 4-1BB or CD28) or those that have enhanced expression of specific checkpoint inhibitors (PD-1, CTLA-4, TIM-3, LAG-3, VISTA), and continuous monitoring of these markers throughout treatment, may lead to more successful tumor control for patients. 

Overall, the manuscript is highly valuable for the scientific community and should be accepted for publication.

We thank the reviewer for his/her positive comments.

======================

Other comments to authors:

1) Please check for typos throughout the manuscript.

Thank you for the kind suggestion. We have re-edited the text looking for typos.

2) Line 388 and further, Figure 7 – with regards to markers of T cell functionality authors are kindly encouraged to cite the following article that extensively covers various aspects focusing on function/dysfunction of T cell-based immunotherapies. DOI: 10.3390/cancers14041078

We have now included the above review as a reference.

Reviewer 2 Report

In the manuscript entitled "Multiagent intratumoral immunotherapy can be effective in tumor clearance and generation of systemic T cell immunity", Gilman et al present extensive testing of different immunotherapy interventions in a mouse lymphoma model. Authors initially test the combination of Poly(I:C) with immunostimulatory cells expressing GM-CSF, but find little effect with these cells specifically, regardless of irradiation/heat-killing of cells. Authors then show that administration of TLR3 agonist Poly(I:C) in combination with anti-TIM3 or OX40-ligand results in the best overall result in terms of mouse survival with tumour challenge. The authors do several FACS experiments on splenic and tumoral immune cell populations, but no significant differences betweeen groups seem to lead to any specific mechanistic insights. 

Overall the authors present a large body of work that seems to be well designed and robustly performed. The data is presented well and the conclusions seem to be appropriate. There is room for improvement in regards to discussion and presentation of some of the data. Following revision to address the concerns listed below, the manuscript should be acceptable for publication. 

Specific concerns:

(1) The title is too vague. As the study is almost all A20 model, this should be included in the title. The abstract should also conclude about lymphoma therapy more specifically. 

(2) In the introduction, authors should talk more about the specific model that was chosen (A20). What is the clinical status of immunomodulatory therapy (such as CPIs) for lymphoma? Has Poly(I:C) been used in lymphoma?

(3) IHC studies are introduced in the methods section but there is no IHC data. 

(4) Throughout the study, authors have used intratumoral delivery for injection of the immunomodulatory agents. Recent work has indicated that Poly(I:C) might work better delivered IV (see http://dx.doi.org/10.1136/jitc-2020-001224). Have authors tried other routes? Authors should provide some introduction to their thinking in the IT route and discuss this in the discussion section. 

(5) Line 235: Authors use the term MHC-disparate, it might be nice to explain the term and the reasoning for this. 

(6) Line 240: Sentence about GM-CSF cells is awkwardly phrased when none of the treatments led to complete regression

(7) Figure 1: The data seems to indicate that Poly(I:C) alone does much of the work. A statistical comparison of poly(I:C) vs Poly(I:C) + other treatments should be provided. 

(8) Line 254: An interpretation/conclusion line such as "This experiment shows..." would be helpful here. 

(9) Line 269: "Next we aimed to [assess whether we could] augment.."

(10) Line 271: "Previous studies by us..." What models were these done in? Provide information in the manuscript. 

(11) Line 284: Provide a summary of these experimental results

(12) Line 302: Provide data on T-cell depletion as supplement

(13) Figure 3: Provide survival graphs similar to other figures

(14) Figure 4: Ideally it would be nice to see Poly(I:C) alone in this experiment, as it is difficult to compare between experiments. 

(15) Line 355: Intratumoral immunotherapy seems to have minimal effects on T cell subsets. Authors should reword 

(16) Figures 5-7: The large panel of antibodies used for analyses is impressive. It is a shame not to see more data from this. Have authors performed a TSne analysis? It would be nice to see if any obvious differences can be seen with dimensional reduction. This would add to the paper nicely. 

(17) Line 409: "Figure M"?

(18)  Line 388-416: This paragraph is very dense and does not seem to come to much of a conclusion. Authors should shorten this and summarize what it means better. 

(19) Figure 7: Authors present the %CD8+IFNg+ or %CD4+IFNg+ of CD45+. Would this not be better presented as the %IFNg of CD8+ or CD4+ respectively, or perhaps IFNg MFI of this gated population? This seems like a strange way to analyze this data. 

(20) Authors should expand on the use of GM-CSF in clinical applications. Has GM-CSF-OV been used in lymphoma specifically?

(21) Given the central importance of Poly(I:C) here, authors should have more discussion of Poly(I:C) and its role in the clinic, and also its potential for lymphoma specifically. 

(22) Authors should make efforts to discuss the findings more generally and place them in context with literature. More references to other studies and less internal Figure references. 

(23) Conclusion statements are too high level and more future facing. Authors should fully rewrite this section to make clear concise conclusions of what the study shows. Only one sentence on next steps/future direction should be used. 

Author Response

In the manuscript entitled "Multiagent intratumoral immunotherapy can be effective in tumor clearance and generation of systemic T cell immunity", Gilman et al present extensive testing of different immunotherapy interventions in a mouse lymphoma model. Authors initially test the combination of Poly(I:C) with immunostimulatory cells expressing GM-CSF, but find little effect with these cells specifically, regardless of irradiation/heat-killing of cells. Authors then show that administration of TLR3 agonist Poly(I:C) in combination with anti-TIM3 or OX40-ligand results in the best overall result in terms of mouse survival with tumour challenge. The authors do several FACS experiments on splenic and tumoral immune cell populations, but no significant differences betweeen groups seem to lead to any specific mechanistic insights. 

Overall the authors present a large body of work that seems to be well designed and robustly performed. The data is presented well and the conclusions seem to be appropriate. There is room for improvement in regards to discussion and presentation of some of the data. Following revision to address the concerns listed below, the manuscript should be acceptable for publication. 

We thank the reviewer for the very thorough evaluation of our manuscript and for her/his thoughtful comments. We have addressed the specific concerns of the reviewer and have significantly edited our manuscript including additional figures as requested.  We address his/her concerns in our point-by-point responses below. 

Specific concerns:

(1) The title is too vague. As the study is almost all A20 model, this should be included in the title. The abstract should also conclude about lymphoma therapy more specifically. 

We have made the title and conclusion more specific by adding A20 lymphoma to the title.

(2) In the introduction, authors should talk more about the specific model that was chosen (A20). What is the clinical status of immunomodulatory therapy (such as CPIs) for lymphoma? Has Poly(I:C) been used in lymphoma?

Our laboratory has extensively studied A20 over the years in response to immunotherapy. Some of our work is now cited in the introduction and discussion. We also discuss the clinical status of CPIs and Poly(I:C) in the introduction.

(3) IHC studies are introduced in the methods section but there is no IHC data. 

The IHC data is captured in Figure 7O-Q. We have added representative images (Figure 7Q) to better highlight the use of this technique in the paper.

(4) Throughout the study, authors have used intratumoral delivery for injection of the immunomodulatory agents. Recent work has indicated that Poly(I:C) might work better delivered IV (see http://dx.doi.org/10.1136/jitc-2020-001224). Have authors tried other routes? Authors should provide some introduction to their thinking in the IT route and discuss this in the discussion section. 

We have some information regarding the reasoning for utilizing intratumoral administration in the introduction section and have added more clarification in the discussion section. The above referenced manuscript is now discussed in our revised manuscript.

(5) Line 235: Authors use the term MHC-disparate, it might be nice to explain the term and the reasoning for this. 

This term is now better described in the discussion.

(6) Line 240: Sentence about GM-CSF cells is awkwardly phrased when none of the treatments led to complete regression

The sentence has been modified to clarify the hypothesis behind the experiment.

(7) Figure 1: The data seems to indicate that Poly(I:C) alone does much of the work. A statistical comparison of poly(I:C) vs Poly(I:C) + other treatments should be provided. 

Thank you for bringing up this point. We have added a sentence stating that the poly(I:C) and poly(I:C)+GM-CSF secreting cells were not different from each other. However, to aid in the ease of visualizing data, only comparisons between PBS controls are included on graphs.

(8) Line 254: An interpretation/conclusion line such as "This experiment shows..." would be helpful here. 

We added a summary sentence to address this concern.

(9) Line 269: "Next we aimed to [assess whether we could] augment."

We have updated this line accordingly.

(10) Line 271: "Previous studies by us..." What models were these done in? Provide information in the manuscript. 

We have added information regarding the use of 12B1 leukemia in BALB/c as the model.

(11) Line 284: Provide a summary of these experimental results

We have added a summary statement of the results stating that heat or irradiation-stress did not augment the anti-tumor response.

(12) Line 302: Provide data on T-cell depletion as supplement

We have added Figure 3 B and D to show T cell depletion data.

(13) Figure 3: Provide survival graphs similar to other figures

We did not include a survival graph here because all mice in each group grew tumor and none survived.

(14) Figure 4: Ideally it would be nice to see Poly(I:C) alone in this experiment, as it is difficult to compare between experiments. 

We appreciate the suggestion here. Unfortunately, we did not collect data from this group during the studies so we are not able to add that into this figure but will consider this for future experiments.

(15) Line 355: Intratumoral immunotherapy seems to have minimal effects on T cell subsets. Authors should reword 

We have reworded our subheading.

(16) Figures 5-7: The large panel of antibodies used for analyses is impressive. It is a shame not to see more data from this. Have authors performed a TSne analysis? It would be nice to see if any obvious differences can be seen with dimensional reduction. This would add to the paper nicely. 

We appreciate this very useful suggestion. Unfortunately, we did not perform tSNE analysis with this data but will consider this for future publications.

(17) Line 409: "Figure M"?

This has been corrected to “Figure 7M”

(18)  Line 388-416: This paragraph is very dense and does not seem to come to much of a conclusion. Authors should shorten this and summarize what it means better. 

We have abbreviated and streamlined this paragraph to discuss only the data that is significantly different.

(19) Figure 7: Authors present the %CD8+IFNg+ or %CD4+IFNg+ of CD45+. Would this not be better presented as the %IFNg of CD8+ or CD4+ respectively, or perhaps IFNg MFI of this gated population? This seems like a strange way to analyze this data. 

Our goal from doing the analysis this way was to illustrate the amount of these T cell populations out of the entire immune cell compartment (rather than within just T cells). We have now included Supplemental Figure 4 to illustrate these markers as a percent of CD4 or CD8 T cells.

(20) Authors should expand on the use of GM-CSF in clinical applications. Has GM-CSF-OV been used in lymphoma specifically?

We have added relevant citations around the use of OVs in lymphoma to the discussion section.

(21) Given the central importance of Poly(I:C) here, authors should have more discussion of Poly(I:C) and its role in the clinic, and also its potential for lymphoma specifically. 

We have added more discussion around Poly(I:C) and lymphoma.

(22) Authors should make efforts to discuss the findings more generally and place them in context with literature. More references to other studies and less internal Figure references. 

We have added additional references and tied them to the findings of this manuscript.

(23) Conclusion statements are too high level and more future facing. Authors should fully rewrite this section to make clear concise conclusions of what the study shows. Only one sentence on next steps/future direction should be used. 

We have updated the conclusions to focus more on those taken from the data and abbreviated those discussion future directions.

Reviewer 3 Report

  • Review was nicely designed and presented well. Methods are clearly explained, detail description provided in general and gradually going into specific details.
  • Each section is demonstrated with adequate details.
  • Appropriate references were cited.
  • Abbreviations elaboration in beginning is recommended to remind the reader in the beginning of the article. 
  • Conclusion part can be more specific with additional detailing with respect to the future directives. 
  • It is recommended to use 'this review or in this work' instead of using 'we' throughout the review.
  • Indent change is recommended for figures to maintain consistency throughout the review with the text.
  • doi is missing for the reference 22.
  • Size of the graphs were too small, increasing the size would help the reader easy to follow.
  • In materials and methods section, it is preferred to show variation for sub headings (example cell culture in line 75) by making them bold to make it obvious to the readers.

Author Response

Review was nicely designed and presented well. Methods are clearly explained, detail description provided in general and gradually going into specific details.

Each section is demonstrated with adequate details.

Appropriate references were cited.

We thank the reviewer for the positive comments.

Abbreviations elaboration in beginning is recommended to remind the reader in the beginning of the article. 

We have attempted to define all abbreviations at the start of the paper to address this concern.

Conclusion part can be more specific with additional detailing with respect to the future directives. We have added information in the discussion and conclusions to address this point.

It is recommended to use 'this review or in this work' instead of using 'we' throughout the review.We have largely rephrased this throughout the paper.

Indent change is recommended for figures to maintain consistency throughout the review with the text.

We have attempted to streamline the indentation but did not want to limit figure sizes to be quite so small while reviewing the manuscript. We will ensure adequate formatting is reached prior to publication.

doi is missing for the reference 22.

There is no DOI associated with this reference.  

Size of the graphs were too small, increasing the size would help the reader easy to follow.

Thank you for this comment. We agree that they are hard to read when small and we will work with the Cancers editorial office to ensure graph sizes are sufficient for publication.

In materials and methods section, it is preferred to show variation for sub headings (example cell culture in line 75) by making them bold to make it obvious to the readers.

We have ensured subheadings are bolded throughout the Materials and Methods section.

Reviewer 4 Report

In the manscript entitled “multiagent intratumoral immunotherapy can be effective in tumor clearance and generation of systemic T cell immunity”, the authors used the A20 murine lymphoma model to evaluate the effect of different combination immunotherapies through intratumoral injection. What they found was that the single treatment with either Poly (I:C) or GM-CSF-secreting allogeneic cells only slightly controlled tumor growth. On the contrary, the combination of Poly (I:C)  with GM-CSF-secreting allogeneic cells resulted in an improved benefit. The same improvement were observed when co-intratumoral injection of anti-Tim-3 blocking antibody, anti-OX40 agonizing antibody and Poly IC. However, the results were not enough to support their conclusion.

1.     In the murine tumor model, they evaluted the efficacy of combination immunotherepies. However, the authors should consider the toxicity of combination therepy, especially three reagents.

2.     In the study, the authors only used only murine A20 lymphoma. They should use some other murine tumor models to make their conclusion general.

3.     In Fig.5, Fig.6 and Fig.7, the authors should provide the level of significance. There were no asterisks in those figures.

4.     In Fig. 3, what was the efficiency of depleting CD4 T cells and/or CD8+  T cells? The authors should provide the data instead of showing “data not shown”.

Author Response

In the manuscript entitled “multiagent intratumoral immunotherapy can be effective in tumor clearance and generation of systemic T cell immunity”, the authors used the A20 murine lymphoma model to evaluate the effect of different combination immunotherapies through intratumoral injection. What they found was that the single treatment with either Poly (I:C) or GM-CSF-secreting allogeneic cells only slightly controlled tumor growth. On the contrary, the combination of Poly (I:C)  with GM-CSF-secreting allogeneic cells resulted in an improved benefit. The same improvement were observed when co-intratumoral injection of anti-Tim-3 blocking antibody, anti-OX40 agonizing antibody and Poly IC. However, the results were not enough to support their conclusion.

We thank the reviewer critically reviewing our manuscript and giving us the opportunity to resubmit with revisions. 

  1. In the murine tumor model, they evaluted the efficacy of combination immunotherepies. However, the authors should consider the toxicity of combination therepy, especially three reagents.

Thank you for this point. We did not observe any side effects or signs of toxicity in any of the treated animals and have included this as a statement in the results section.  

  1. In the study, the authors only used only murine A20 lymphoma. They should use some other murine tumor models to make their conclusion general.

Figure S1 utilized the B16 melanoma model and showed that the combination of immunotherapy slowed tumor growth but was not as successful as the A20 tumors.

  1. In Fig.5, Fig.6 and Fig.7, the authors should provide the level of significance. There were no asterisks in those figures.

All statistically significant items are listed on figures 5, 6 and 7. No asterisks are included on certain graphs because no statistical significance was reached for those particular results. This is likely due to the large variability seen across samples within groups, as no outliers were excluded from results.

  1. In Fig. 3, what was the efficiency of depleting CD4 T cells and/or CD8+  T cells? The authors should provide the data instead of showing “data not shown”.

This data has now been included as Figure 3 B and D.

Round 2

Reviewer 2 Report

Authors have adequately addressed concerns raised in the first round of review.  

Reviewer 4 Report

The manuscript has not been significantly improved to warrant publication in cancers